# The Body after Cancer: A Qualitative Study on Breast Cancer Survivors’ Body Representation

**DOI:** 10.3390/ijerph191912515

**Published:** 2022-09-30

**Authors:** Valeria Sebri, Ilaria Durosini, Davide Mazzoni, Gabriella Pravettoni

**Affiliations:** 1Department of Oncology and Hemato-Oncology, University of Milan, 20122 Milan, Italy; 2Applied Research Division for Cognitive and Psychological Science, IEO, European Institute of Oncology IRCCS, 20141 Milan, Italy

**Keywords:** breast cancer, body image, psychological intervention, breast cancer survivors, emotions

## Abstract

Objective: The relationship with the body is a relevant issue for breast cancer survivors. Oncological treatments damage their bodies due to scars, weight gain, and other side effects. Starting from the efficacy of psychological interventions for breast cancer survivors, a tailored psychological support program was provided to promote overall well-being after illness dealing with bodily signals and related emotions and thoughts. This study presents changes in the description of the relationship with their bodies as well as participants’ emotions and thoughts before and after a psychological intervention. Methods and Measures: Eighteen women answered questions related to their bodies before and after the psychological intervention. Results were analyzed in accordance with the procedure of the Word Association Analysis through the T-Lab software and the Qualitative Thematic Analysis. Results: Participants reported a great awareness of their bodies and the desire to take care of them daily. In particular, the body is now perceived as a helper to sustain breast cancer survivors in their everyday activities. Conclusion: The words and the themes that characterized the participants’ reports highlighted the impact of cancer diagnosis and oncological therapies on breast cancer survivors. The participation in the psychological intervention focused on self-compassion towards their body helps women to create an improved body perception.

## 1. Introduction

Breast cancer diagnosis and treatments expose patients and survivors to physical and psychological consequences, until years after diagnosis and treatments. On a physical level, breast cancer survivors have to deal with possible long-term side effects of oncological care (e.g., fatigue, vomiting, pain, and risk of infections). On a psychological level, diagnosis, changes in lifestyle and social relationships, and detrimental sexual functioning often increase negative emotions, such as emotional distress, depression, and anxiety [1,2]. The discrepancy between women’s current and ideal self-image generally leads to dissatisfaction and emotional distress, promoting worry about physical appearance and the belief that others evaluate their bodies continually [2].

Specifically related to the body, studies demonstrated that the experience of breast cancer seriously infringes survivors’ body image (BI) [3,4]. As a definition, BI is conceptualized as an “internal representation of one’s own outer appearance” [5]. It is at the core of bodily self and involves the mental representation of one’s own body and related emotions [6]. Thus, BI can affect several dimensions: perceptions (i.e., accuracy of estimated body size), attitudes (i.e., subjective satisfaction about one’s own body), cognitions (i.e., involvement in appearance belief about the body), sensations and emotions, and behaviors (i.e., compensatory behaviors, such as dieting and physical activity) [7]. The removals of breast(s) as well as other bodily effects (e.g., hair loss, skin discoloration, and visible scars) impair their subjective experience within one’s own BI and related emotions and thoughts [8]. As a result, breast cancer survivors tend to be severely worried about their physical appearance and develop the belief that others observe and evaluate their bodies continually [9]. The discrepancy between one’s own current and desired self-representations leads to feelings of emotional distress and dissatisfaction, according to theories of self-objectification [10] and following the Self Discrepancy Theory by Higgins [11]. In this regard, the experience of femininity and sexuality after illness depends not only on oncological treatments and their consequences, but also on the intrapsychic negotiation within their relational context [12]. Additionally, breast cancer survivors may experience changes in their identity and in their body appearance, even perceived as a source of danger and fear. Firstly, after diagnosis, women could reframe their identity as that of a patient, with notable consequences on their life activities and self-management [13,14]. Implications of their changed self-identity could lead them to the perception of being women at risk, unable to manage their routine and relationships [15]. On the other side, breast cancer survivors are worried about the possibility of perceiving inner sensations, which are often associated with the fear of cancer recurrence. As consequence, the oncological experiences can damage the mental representation of the body of breast cancer survivors, with a negative impact on emotions and the overall sense of bodily self (i.e., body image) [5,6,16]. Accordingly, breast cancer survivors have to deal with a renovated Self that integrates a new self-representation (namely Injured Self), which is based on the autobiographical memory of the oncological experience [17]. Thus, it is of paramount importance to support breast cancer survivors and the relationship with their body after illness aiming promoting a positive experience with their inner sensations and related emotions. Addressing bodily issues could also help them to re-define their life after breast cancer and overcome the emotional trauma related to illness [18].

### Psychological Interventions on Self-Compassion for Breast Cancer Survivors

Regarding the contents of psychological interventions, current studies show the importance of the promotion of resources aiming at dealing with personal issues by treating oneself and the others with kindness and support, which is known as self-compassion. Interest in self-compassion interventions for breast cancer survivors has grown significantly over years [19]. As a definition, compassion is “a sensitivity to suffering in self and others with a commitment to try to alleviate and prevent it” [20]. Closely related to mindfulness [21], it is characterized by six attributes as follows: “Sensitivity” (the ability in perceiving others’ emotions), “Sympathy” (showing concerns for the other person’s suffering), ”Empathy” (that is the capacity to feel other people’s emotions), “Motivation” (in terms of act or response toward the suffering that others express), and ”Distress tolerance” (when managing difficult situation without overwhelmed feeling and with a non-judgment attitude [22]. In this way, self-compassion intervention can decrease anxiety and depression and promote quality of life.

Nowadays, studies show that several psychological interventions on self-compassion are helpful for breast cancer survivors [23,24,25]. The overall aims are based on the promotion of emotion regulation and abilities to sustain and enrich valued lifegoals [22]. Moreover, self-compassion intervention helps people to recognize the suffering of others and of oneself as well as to tolerate the distress by learning new ways of managing emotions [23]. At the same time, online interventions are gaining increased popularity (also during the COVID-19 pandemic and lockdown) as a cost-effective way to address these persistent challenges, using technologies to stay connected with others [26,27,28]. Current studies showed the effectiveness of online interventions on BI by enhancing Quality of life and decreasing fatigue [29]. More specifically, on a psychological level, Esplen and colleagues [30] showed the effectiveness of 8-week online text-based group intervention developed by a therapist-led in-person and evaluated in a randomized controlled trial to restore BI in terms of self-schemas and emotional reactions after cancer (ReBIC). Additionally, participants can complete online training sessions at their own convenience, with the added benefit of reviewing sessions of delivered information as often as needed [26]. The presence of other women with similar illness experiences can be an important aspect to promote the engagement, the motivation and the adherence of participants. Indeed, recent studies highlighted that personal motivation can change over time and group support could represent an important aspect that allows women to share their life experiences in a protected context with an expert psychotherapist [31,32].

Despite the theory and evidence sustaining a positive impact of self-compassion and online interventions on BI, the specific changes in bodily representations in breast cancer survivors are still unclear. Starting from this theoretical background, the present study aimed to qualitatively explore the bodily representations on a cognitive and emotional level in breast cancer survivors. A qualitative lexicometric analysis and a thematic analysis were carried out to assess the relationship with their body and related emotions and thoughts in an online psychological intervention focused on self-compassion.

## 2. Materials and Methods

### 2.1. Institutional Review Board Statement

This study was performed in line with the principles of the Declaration of Helsinki and approved by the Institutional Review Board (or Ethics Committee) of the European Institute of Oncology, IRCCS (n. R1598/21-IEO1702).

### 2.2. Participants

Twenty women with a history of breast cancer voluntarily agree to participate in this study. All of them were involved in a psychological intervention focused on body image after cancer using a self-compassion approach. The psychological intervention was carried out in March 2022. All the participants met the following inclusion criteria: (1) women who are 18 and older years old (2) participants who received a diagnosis of breast cancer in the past years, and (3) women who are not under oncological treatment currently. Participants who showed cognitive impairment, inability to understand the study or to sign the informed consent, and/or mental disorders that prohibited their participation to the study were excluded from this research.

A total of eighteen women responded to open questions both at the beginning and at the end of the psychological intervention (age range: 38–69; *M_age_* = 50.6; *SD_age_* = 8.97). The majority of participants had obtained a bachelor’s degree (50%), lived in the North of Italy (83.2%), and worked as white-collar employers (83.2%). Additionally, more than half of them is not currently involved in individual psychological therapy (55.6%), had a partner (72.3%) and one or more sons (55.6%).

### 2.3. Psychological Intervention

Women voluntarily agreed to participate in a 1-month psychological intervention for a total of 4 sessions. Online psychological group was based on 2 h-sessions weekly and was conducted by an expert psycho-oncologist with extensive professional experience in BI issues. Contents of the psychological intervention were based on previous studies focused on BI issues [6,33,34] and, in particular, on the steps of the BI workbook by Cash [35,36]. Accordingly, the main goals were the promotion of positive bodily representations by promoting cognition, emotions, and behaviors. Starting from psychological theories and validated group interventions on self-compassion among different populations [37], psychological program was focused on: (1) the relationship with the body before and after diagnosis and oncological treatments, (2) barriers and facilitators in the body relationship, (3) participants’ BI and the related emotions, and (4) how to manage bodily issues in the future (see more details in Table 1). All breast cancer survivors shared their illness experiences in the online group by exploring their personal bodily representations and related emotions and thoughts within the intervention. Program included discussion, in-group activities, and take-home tasks. For example, participants collected information related to their emotions and behaviors in a diary. All breast cancer survivors participated in each session of the program.

### 2.4. Procedure

Breast cancer survivors received the invitation to participate in the psychological sessions via a mailing list. Meanwhile, a self-selection method was applied to recruit participants via social networks (e.g., Facebook), inviting breast cancer survivors to participate in the project. After the acceptance to participate, they received an email with the information sheet and informed consent form. Participants who were involved in this study met the following inclusion criteria: (1) 18 and older years old (2) having received a diagnosis of breast cancer in the past years (at least five years ago), (3) absence of oncological treatment ongoing, and (4) understanding and speaking of Italian language. Moreover, breast cancer survivors who showed physical or psychological impairments that prohibited their participation in the study, such as inability to understand the study or to sign the informed consent, were excluded from this research. If appropriate for this study, participants received a ZOOM link for the online intervention and were asked to respond to three open questions. Specifically, they were asked to describe: (1) the relationship with their body, (2) the emotional experience related to their body after cancer, and (3) their thoughts about it. The qualitative study was based on prompts designed to elicit the free expression of patients’ thoughts and emotions about their body. There was no word limit or time restriction on this study. Patients were invited to describe their experience freely, in a narrative way. All the answers were completed individually by each participant two times (before and after the psychological intervention). Personal information was deleted to guarantee anonymity. Basic demographic data (e.g., age and type of cancer) were also collected. All the textual answers were collected through Qualtrics platform and analyzed by the research team involved in this study.

### 2.5. Data Analysis

According to the study aims, the analytical process followed two main phases. The first phase was aimed at identifying the more frequent words that the participants used to speak about their own body, before and after the intervention. We thus conducted a lexicometric analysis with the software T-LAB Plus 2021 [38]. This software represents a set of linguistic and statistical tools that allows the automatic analysis of patterns of words and themes related to the explored topic. The mixed-method ability of this software allows the user to perform a set of statistical and linguistic analyses on textual data [39]. More specifically, the corpus under analysis was made by all the words spontaneously used by participants. With the Word Association analysis, it was possible to identify the words that were more frequently used in association with (i.e., in the same elementary context with) the word “body”.

In the present study, this analysis enabled us to detect the words used most recurrently in association with the key-term “body”, before and after the psychological intervention. We analyzed the words spontaneously used by the participants and that more frequently co-occurred with the keyword ‘‘body’’. This allows us to deeper understand participants’ aspects and changes over time.

In the second phase of analysis, we conducted a qualitative thematic analysis with a bottom-up approach [40], allowing the analysis of the patients’ experience and feelings related to their body. Adopting an inductive approach, we did not try to fit the data into a pre-existing frame, but, starting from three main themes (i.e., relationship, emotions, and thoughts towards their body), subthemes were identified through a “bottom-up” process [41]. As described by Braun and Clarke [40], the analytical process followed the main phases of qualitative thematic analysis, as follows: firstly, two authors (V.S. and I.D.) read each text-word many times to familiarize with and understand the contents. Then, an initial coding was referred to the data to identify segments of the textual reports (semantic content). This way, researchers considered a single segment when the answer reported a single main bodily relationship/emotion/thought. Third, the different codes were categorized into potential main themes and sub-themes. Fourth, the thematic map was reviewed by two authors (V.S. and I.D.) and possible discrepancies were solved through discussion. Finally, the sub-themes were labeled. This provided an explicit and defined clarification of the contents. One author (D.M.), who did not take part in the previous phases of analysis, reviewed the coding process and the identified sub-themes to further validate the thematic map.

## 3. Results

The length of the answers was heterogeneous, ranging from a few-word statements to a paragraph of 124 words. In order to understand women’s feelings, thoughts, and the relationship with their body, a Word Association Analysis was conducted through T-LAB. The results highlight qualitative changes in the women’s relationship of their body before and after group sessions, suggesting increasing attention, appreciation, and care for it.

In the first data collection (before the intervention) participants associated with “body” terms related to the self-assessment of their symptoms (such as, “seek”, “observe”), and with aspects related to their negative feelings about it (such as, “uneasy”). For example, women wrote that: “I do not like my body and I do not want to observe myself in the mirror” or “I have a feeling of rejection, and whatever it is that makes me uncomfortable and uneasy with myself” (Table 2, Figure 1). Meanwhile, after the intervention, participants tended to associate the keyword “body” with terms of “care” and “cure”, for example: “I am trying to take care of my body now”. This is supported by the association of the term “body” with words related to “positive”, “pleasure” and “relationships”, who suggested the tendency of women to stay in contact with their new body perception, “looking” after a long time with the signs of oncological treatments. In addition, only after their participation in the psychological group, women associated the word “relax” with the word “body”. For example, a participant reported that: “I feel my body relaxed and stable, well rooted on the ground and in its support surface. I feel my body as an empty and inert container’’ (Table 2, Figure 2).

Additionally, a thematic analysis was conducted on the whole data. The results were grouped into three main themes related to women’s relationship, emotions and thoughts on their body before and after the psychological intervention. Specifically, we identified: 9 sub-themes were in the relationship with the body theme (5 sub-themes for the first data collection and 4 sub-themes for the data collected after the intervention), 10 sub-themes in the emotions theme (6 sub-themes relate to the first data collection and 4 sub-themes for the second data collection), and 7 sub-themes in the thoughts theme (3 for the first data collection and 4 for the second data collection; Table 3). Quotations were included in this manuscript using participants’ ID numbers.

### 3.1. The Relationship with the Body

The first area explored is related to the relationship that women have with their bodies. Before the psychological intervention, women involved in this study described their relationship with their bodies in different ways. Seven women stated an *un-acceptance of their body* due to the physical changes after the oncological diagnosis. The signs of surgery and oncological treatments prevent them from returning to a positive image of themselves. Scarring and weight gain led women to see their bodies like a stranger, developing the desire to return to their bodies before the diagnosis. For example, a woman with a history of breast cancer stated that “I do not have a good relationship with my body. I do not like myself anymore. I would like to lose weight and be toned again. I want to return to my body before cancer (ID6)”. Additionally, a woman with a similar history of cancer reported that “the scar is ugly [...] The aesthetic treatment that I did after the surgery did not give the expected results. The relationship with my body is bad’’ (ID9). In accordance with this, another participant claimed a discomfort relationship towards her body due to the aesthetic appearance and the negative physical sensations perceived: “I don’t like my body. I need to tone my muscles; I have been out of training for many years’’ (ID2). Sometimes, the “signs of cancer” lead a woman to take *distance from her body,* avoiding looking and touching her necked body. She reported that “I do not like my body and I do not want to observe myself in the mirror. When I look at myself in the mirror, it is as if I saw the body of another woman. In particular, I avoid touching and looking at my breasts (ID1)”. In accordance with this, the changed body represents for a participant a clear reminder of her oncological diagnosis (*body as a cancer-related reminder*) and hinder the process of acceptance of the disease. A woman reported: “The sight of my naked body disturbs me: this happens not so much for the brutality of the image, but because I am forced to rethink the lived experience” (ID12). Another sub-theme is related to the *“insecure attachment” towards the body*. The psychologists Bowlby [42] used the word “attachment” to refer to the quality of mother-child attachment relationship that people are able to form in their life. Generally, this relationship can be grouped as either secure or insecure. In an insecure relationship, which is a type of the mother-and-child bond, the infant is insecure in the presence of the caregiver. The child resists staying in contact with it and is wary of the stranger. In this study, we propose to employ the expression “insecure attachment” to highlight the relationship between cancer survivors and their bodies after the diagnosis. Specifically, some women highlighted that while they tend to love their body, they harbor the fear that it may somehow hurt and betray them again. One woman stated that “I love my body, I talk to my body often and ask it to support me, not to fight me that we are one. I know I have a strong connection with him, I hope the body listens to me (ID10)”. Despite this, two women are able to maintain a *collaborative relationship with their bodies* after the cancer diagnosis, reporting their thoughts as follows: “My body, when dressed, is like before illness. So, I don’t have a bad relationship with it. I just see myself a little fat after the surgeries, also because I had sharply increased the amount of food at meals. I wanted to fight against cancer. I also fought in this way. So, I would like to recover in better physical shape if I can. But without worries” (ID11) and “My body is not perfect, but it’s still better than it could be after cancer. I am not still in peace with the idea that it is changed, and I will not have a baby in the future, that I will no longer feel sexual pleasure as before illness, and that I feel physical pain that should not be felt at my age. Nevertheless, I accept it, in exchange for health stability that allows me to survive (ID13).

After the psychological intervention, participants reported different descriptions of the relationship with their body, compared to the initial data collection. Even if three women continue to describe a difficult relationship with their body *(un-acceptance of their body)* due to the psychological impact of the evident signs of the therapies (e.g., “The relationship with my body is very difficult. I can not accept the changes that the disease and its treatments caused to my appearance” (ID4), the majority of participants highlighted positive changes in the relationship with their body after the psychological intervention. For example, seven women involved in this research highlighted the tendency to maintain a *collaborative relationship with their bodies* despite the signs of the diagnosis. After the psychological intervention, a greater number of participants compared to the initial phase of the study highlighted aspects of their relationship with the body like “I have a good relationship with my body, over time I have learned to respect its new rhythms (ID12)”, “I accept my body, I would like to improve some things, but overall I accept myself (ID7)”, and “I have a collaborative relationship with my body, I try to take care of it as much as possible. Sometimes I listen too much to its signals and this in some cases is not good, but I am aware of it and I am working to try to find a balance” (ID10). This is in line with the strong desire to improve the *confidence with the body,* recognizing their capacity to overcome the sign of the diagnosis. In some cases, women recognize that their body “deserves more attention and positive stimulation from me… In all its parts… Especially the parts that I consciously ignore because I feel that they are foreign parts that inhabit me (e.g., prosthesis). I have to reconcile myself with all the aches and pains (ID13)”. Interestingly, after the psychological intervention, participants who described a tendency to avoid looking at and touching their body after the diagnosis expressed the desire to *stay in contact with the body* (“I decided to look my body in the mirror when I am naked. Despite the fear related to the view of ugly scars, I want to take courage and look at myself every day... but it is not easy (ID8)”).

### 3.2. Emotions Related to the Body

The second area explored by participants is the emotions related to their bodies. Before psychological intervention, a participant reported a *sense of alienation* towards their bodies. In other words, she stated a sense of being in a body that does not belong to her and that is not under control. The actual body is often compared with the body before the surgery and oncological treatments, which leads to difficulties in the psychological acceptance of bodily changes. In line with this, a breast cancer survivor reported that: “Sometimes I feel a sense of alienation: this is my body right now...but I no longer recognize it as mine, it does not represent myself...it is as if I were looking at my body from the outside. Sometimes, I feel frustrated because I have no control over my body: no matter how hard I try, I could not come back to my previous weight; as much as I try, my mental brilliance and memory are not those of the past; No matter how hard I try, the physical energies and abilities are not comparable to those before breast cancer (ID13)”. In accordance with this, another participant claimed that *body signals are perceived as unclear.* This woman reported difficulties to recognize and understand bodily signals, as follows: “The signals of the body are a burden for me! Maybe they try to give me some information about myself...but I do not receive these signals, I am not able to understand and interpret them!” (ID14). Moreover, the body is sometimes a clear reminder of the history of cancer, leading to the promotion of the *fear of cancer recurrence* (“I am sometimes scared…not every day… I am afraid when I am forced to think about what I have experienced. Right now, I feel like my body’s held up” (ID11). Accordingly, five women expressed *negative emotions* of rejection, discomfort, and contempt towards their body. For example, a woman stated: “I have a feeling of rejection, and whatever it is that makes me uncomfortable and uneasy with myself” (ID1). Similarly, another participant reported: “I feel a feeling of contempt toward my body...when I am naked, I avoid looking in the mirror as much as possible” (ID4). In other cases, four women experienced *ambivalent emotions* related to their body at the same time, ranging from positive thoughts concerning their physical appearance to negative emotions related to physical changes and side effects of oncological treatments. On the one side, it is possible that women feel positive emotions about the body, appreciating some parts of it; on the other side, the same participants consider their body as a source of negative feelings and evaluations, as follows: “Several times I feel good inside my body, I look beautiful and this makes me feel good...other times I see myself bigger and this makes me feel awkward” (ID3) and “Searching for a balance with my body. My body responds well to therapies, and this is a (great) gift that makes me. On the other hand, I must accept and counteract all the side effects of oncological treatments, both on a physical and psychological level” (ID13). Despite this, three participants are able to emphasize positive feelings about their body, appreciating the progress made over time, experiencing the sensation of being in a peaceful state, and feeling a *sense of gratitude* towards it for the progress made after breast cancer (“Sometimes, I appreciate my body because “it could be much worse”...instead it gives me the opportunity to be still alive and live an almost normal life: I can do everything, with very few limitations and especially without help” (ID13), “My body has changed and has many scars. I look at them every day and I am sorry that my body has had to undergo amputations...but I know that there are just bodily changes that I have to accept...however, it is strange, sometimes I do not recognize myself!” (ID10), and “I feel a sense of gratitude because, despite the oncological therapies, my body has been able to recover from the illness and allowed me to have a child” (ID12).

After the psychological intervention, participants’ emotions towards their body slightly changed. Although a woman still reported a *sense of alienation* towards her body (“I have no emotions...I am detached from my body...the mind wanders off on its own without relating to the body...” (ID14), the majority of participants perceived many emotions related to the body after intervention. Specifically, three women maintain *negative emotions* towards their bodies, even if different in comparison to emotions perceived before intervention. They indeed reported emotions such as anger, despair, discomfort, or sadness due to the signs of the oncological therapies, which are still visible on their bodies (“Speaking on my body, I feel almost exclusively negative emotions, such as, anger, discomfort, and sadness” (ID4) and “I feel discomfort for the physical pain. The ongoing hormonal treatment damages my poor bones every day!” (ID9). Moreover, one participant recognized a sense of guilt towards her body for neglecting it for a long time. This is a new emotion that was not reported before intervention. Specifically, she claimed: “I realize that I have neglected it because of the illness for a long time” (ID15). At the same time, four participants reported *ambiguous emotions*, like “love and hate” for their bodies. Interestingly, after the psychological intervention, the emotional swing is related to the process of integrating a new image of themselves, as follows: “Sometimes I like my body a lot, sometimes I just don’t: I find myself swollen, fat, and I think that I could attract more positive attention with a more beautiful body” (ID3) and “I feel an alternation of joy and inconstancy…. I would care more about a healthy diet, but my head does not give attention to this point!!” (ID5). Contrarily, four participants emphasized a strong *sense of gratitude* to their body for being able to bear the weight of the oncological treatments. For example, they reported: “In this moment of personal reflection, I feel as if I were grateful to it for having supported surgical operations, for having resumed, after the time of convalescence, to move, to act, to support me in daily activities. Despite the wounds-which are quite extensive-my body did not stop: it adhered not only to the demands of daily life but also to the desire for movement, following my cultural and social interests. So, I’m very grateful!” (ID11), and “I feel a sense of gratitude because my body has been able to endure all the oncological interventions and treatments...I can count on it every day” (ID12)”. In line with this, a new awareness related to the body is reported: “I am grateful to my body for not betraying me, despite the wounds-they are many–inflicted…but I am also aware of the fact that I neglected it, as if it were forced to follow me. I didn’t have much respect for my body over illness: I asked him a lot. For example, I treated only the face: I always attributed a greater value to the face. The face is the expression of the inner world: therefore, for me it has always been more important. Now I think that the whole body has equal dignity (ID11)”. Interestingly, the perception of *unclear bodily signals* as well as *fear of cancer recurrence* disappeared after psychological intervention.

### 3.3. Toughts Related to the Body

Finally, the third area was related to the thoughts towards the body. At the beginning of the intervention, four participants reported high levels of *cognitive burden* due to fatigue (e.g., “I don’t feel my body at the top. I have physical energy but then I feel out of training. I am very tired!” (ID3)) and a sense of body rejection by saying: “The first thought that comes to my mind is indifference and rejection towards my body. When I have to think of my body, I perceive a mental fatigue” (ID1) and “Obviously I would have preferred not to have to run into this disease, I look ugly, and I have to deal with age but also with the consequences of cancer...” (ID8). Breast cancer is indeed reported as a strong and vivid experience that affects well-being daily, leading to a *sense of impotence.* For example, they stated: “The first thought is that I would like to go be like before the disease experience...but I know that this is not possible...so I feel helpless” (ID4) and “I would like to help both my body and me to feel good, calm, full of energy...I would like to have a healthy diet...but my tiredness is too high...then, anxiety affects my body negatively” (ID14). On the contrary, four participants reported that their main thoughts were focused on *taking care and time* for their body daily. For instance, a woman spoke about the meditation practice as a way to listen to her body sensations (“I often dedicate myself to meditation and relaxation, it helps me to know myself and my body” (ID6)), which is in line with the aim of learning how to respect the body (“I know I should definitely respect my body, without stressing it every day” (ID12)). Interestingly on this topic, the ability to stay in contact with the body and its related sensations may sometimes become an issue to solve. In particular, a breast cancer survivor stated: “I should not listen to my body too much! I tend to stress about it! Sometimes, my body tends to make a distance from me; sometimes, I am too focused on any physical changes and inner sensations. This stresses me a lot!” (ID10).

At the end of the psychological intervention, participants reported the usual sensations of fatigue and tiredness related to a *cognitive burden* (“I feel my body tired because of my everyday issues” (ID7)) and a *sense of impotence* as before intervention. On the contrary, six women reported their desire *to take care and time* for their bodies. For example, women spoke about their involvement in relaxation practices and positive thoughts of caring and love towards their bodies, as follows: “I have the desire of a relaxing massage to fight physical tensions and relax the mind” (ID9), “My body is beautiful, strong and, at the same time, delicate. I have to take care of it every day” (ID12), and “Going forward: I have so much to do and want to do it. Looking forward and leaving behind everything that makes me too heavy. I want to be more serene and happy” (ID13). Interestingly, a stronger awareness about the importance of caring their bodies to sustain physical and psychological well-being emerged, for instance: ‘‘Now I know that relaxation and serenity help me to keep my negative thoughts under control” (ID5) and “I know that I have to take better care of it by physical training regularly, drinking more water, and relaxing from working tensions” (ID13) and “The relationship with my body is, at this time, overall positive. I do not feel any tension. However, I do not abandon the awareness of improving my physical appearance. Now I want to dedicate myself to the care of my body” (ID11). Finally, four women reported a new point of view regarding their bodies. Interestingly, a woman defined her *body as a helper*, as follows: “My body is a friend, not an enemy! It suffered with me, it felt pain, but it never gave up. We will never give up. I will love it more than I have in recent years (ID8)”, aiming at making a new positive relationship (“I want to thank by body for its efforts over cancer illness, trying to do not stressing it anymore” (ID15), “I think it is important to have a positive relationship with my body. It allows me to live on this earth! I should listen to my body and take care of it as much as possible” (ID10)), and “My first thought is compassion. I suffer with my body, and we react together: mind and flesh. We react to therapies, side effects, mood swings, fatigue, and everyday stresses that arise from work, home, and family care. After all, yes, I perceive it as my ally, that despite everything still gives me satisfaction and leads me to look” (ID13).

## 4. Discussion

In this contribution, we describe the process of change that occurred in a group of breast cancer survivors over an intervention based on self-compassion towards their body. More specifically, in this study, we focused on the process of change in the relationship, emotions and thoughts on their body that occurred in a group of breast cancer survivors over an online psychological intervention. The lexicometric analysis of the participants’ answers suggested a general increase in the use of words communicating attention, appreciation, and care toward the body. This tendency was further confirmed in the analysis of the answers through qualitative thematic analysis, which identified twenty-six sub-themes in reference to the three main themes proposed.

Firstly, the results emphasize an increasingly positive relationship with the body, in terms of increased acceptance and less distance, more confidence and collaboration. Literature shows that specific psychological intervention (e.g., cognitive behavioral program) can result in body acceptance thanks to short-term reductions of bodily issues, such as weight-stigma in people with obesity [43]. Positive implications could be obtained also in the satisfaction with relationship and the perception of being attractive, which lead to a general well-being and positive emotions [44]. This is in line with emotional improvements obtained at the end of psychological intervention. Specifically, we observed a reduction in the frequency and in the intensity of the reported negative emotions. Accordingly, current studies highlight the positive impact of psychological intervention focused on self-care and kindness on BI in breast cancer survivors [6]. In particular, Lewis-Smith and colleagues demonstrated that this kind of intervention can promote body appreciation and acceptance, reducing weight and shape concerns [6]. Moreover, improvements in self-esteem, which is strongly associated with BI issues in a breast cancer population, play a relevant role too [2]. At the same time, it is possible to see that some participants still reported negative aspects of their relationship with the body even after the intervention. In this regard, we must recognize that the past experience of breast cancer could have seriously impacted on these dimensions [3,4], in a way that cannot be completely healed by this brief psychological intervention. In this regard, literature shows that depression and anxiety are some of the main emotional issues with long-term effects. A study by Breidenbach and colleagues demonstrated that breast cancer survivors had higher levels of depression also five to six years after diagnosis [45]. Having children and lower vocational status were correlated with depression, in particular; on the other hand, surgery type was associated with anxiety, whereas age and comorbidities were predictors of both depression and anxiety symptoms. Moreover, increased levels of depression were observed in women having children and comorbidities, while more anxiety was likely after cancer recurrence. However, in all the cases, it was possible at least to observe some indices of an increased consciousness about the importance of a good relationship with their body. Indeed, we observed more awareness about the importance of the body to promote a general well-being and an increased agency, which was expressed through a less sense of impotence. Coherently, a study by Mifsud and colleagues demonstrated that a brief self-compassion intervention aiming at addressing BI issues can promote breast cancer survivors’ well-being thanks to their awareness of their oncological experience and the related responses in a broader context [46]. Additionally, an interesting result is referring to a new body perception, which is strictly related to the core of the intervention. At the beginning, women stated that they would not touch and look at their body in the mirror. On the contrary, the desire to make contact with bodily parts, even changed due to oncological surgeries and treatments (e.g., breast), emerged at the end of the psychological sessions. In line with this, a new relationship with the body after breast reconstruction could be expressed by choosing to counteract negative self-images with artistic and meaningful tattoos in order to sustain body acceptance [47]. Lastly, healthy activities (e.g., relaxation and physical exercises) were also considered by participants as positive in order to promote well-being and improve their abilities to manage their emotions. This result is in line with literature that suggested that sport and physical activity could represent an important aspect after cancer, helping women to improve cognitive abilities and quality of life [14,16,48].

### Limitations and Future Studies

One limitation has to do with the study design, which did not include a control group nor a quantitative evaluation of the intervention efficacy with validated measures. Our results thus shed light on the process of bodily representations change during the intervention, rather than statistically demonstrate the efficacy of the intervention in comparison with alternative approaches. Some cautions should be adopted in generalizing the results to other cancer and/or healthy populations. Thus, the present paper contributed to understanding the relevance of a body-compassion intervention, which need to be further explored in order to add richness of healthcare professionals’ knowledge. A second limitation has to do with participants’ selection. In fact, it is possible that the participants who decided to participate in the intervention had already a specific attention and interest for their body and its changes. This is in line with the fact that almost half of this sample is involved in individual psychological sessions. Finally, the sample size needs to be increased in order to assess the efficacy of the intervention and generalized results. Moreover, a larger sample size may lead to the possibility of analyzing differences between subgroups in reference to socio-demographic data, for example regarding women who live at the North or South of Italy.

Despite these limits, this study has some important implications at theoretical, methodological, and practical levels. First, at a theoretical level, this study is the first investigating the specific changes in bodily representations of breast cancer survivors, after an intervention which focused on self-compassion towards their body and was delivered in an online setting. As previous research demonstrated, interventions on self-compassion have been conducted with breast cancer survivors, with promising results [23,24,25]. The present study builds on this evidence, proposing an online intervention which is as a cost-effective way to conduct psychological interventions and to promote interaction between breast cancer survivors overcoming space and physical barriers. Second, this study adopts two complementary approaches to text analysis. The first one, lexicometric, allowed us to identify, with a quantitative tool, the specific words that constituted the body perception, before and after the intervention [38]. The second one, thematic analysis [40] is a more strictly qualitative approach that allows us to deeply describe the breast cancer survivors’ relationship with their body, their related cognitions and emotions. These two approaches (lexicometric and qualitative) to text analysis have been rarely applied in the same study (for rare exceptions see: [49,50]), but we strongly belief-as this study demonstrated-that they can be usefully integrated to obtain a more detailed and consistent picture of the texts’ meaning. Third, at a more practical level, this study highlights the relevance of psychotherapy groups to improving patients’ confidence in their body. More specifically, this study emphasized the importance of implementing psychological interventions focused on cancer survivors’ needs, describing specific changes that helped the participants to rethink their body, and to promote their well-being.

Future studies could deeper evaluate the efficacy of the psychological intervention with validated measures. Additionally, the inclusion of a control group may demonstrate the efficacy of this specific intervention compared to alternative approaches. Moreover, it could be interesting to explore the changes in bodily perceptions in patients with other kinds of chronic diseases. As an implication, this study could be helpful to increase knowledge about body-compassion for health professionals. Future psychological interventions could assess and address body-compassion in order to promote well-being in breast cancer survivors, involving related emotions and thoughts.

## 5. Conclusions

This study aimed at describing the relationship of a group of women with a history of breast cancer who took part in an online psychological intervention with their body, emotions, and thoughts. The words and the themes that characterized the participants’ reports were analyzed. Most participants reported that the breast cancer experiences, with the related therapies, had a negative impact on their relationship with their body, their emotions, and thoughts. This psychological intervention helped women to improve the quality of such relationships, meeting the participants’ needs for rebuilding contact, confidence, and collaboration with their body. At an emotional level, the intervention promoted an elaboration process, which helped to reduce the intensity of negative emotions related to the illness experience and improve the sense of gratitude towards their body. Finally, at a cognitive level, the reduction in a sense of impotence to do something to return to their bodies as before the illness and a sense of increased power and vitality emerged, perceiving the body as a helper that can sustain women in their daily life activities.

## Figures and Tables

**Figure 1 ijerph-19-12515-f001:**
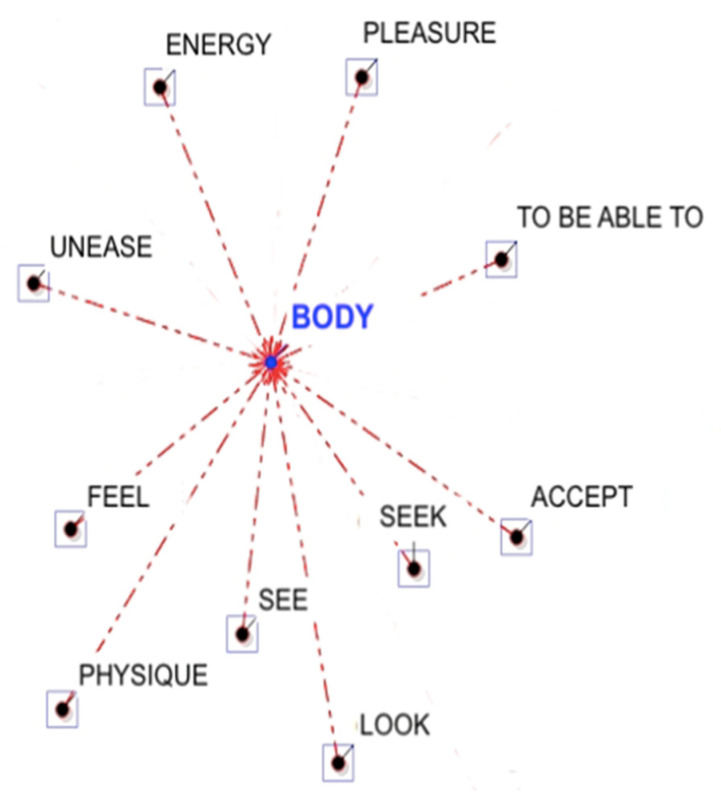
Graphical representation of the Word Association Analysis (before the psychological intervention).

**Figure 2 ijerph-19-12515-f002:**
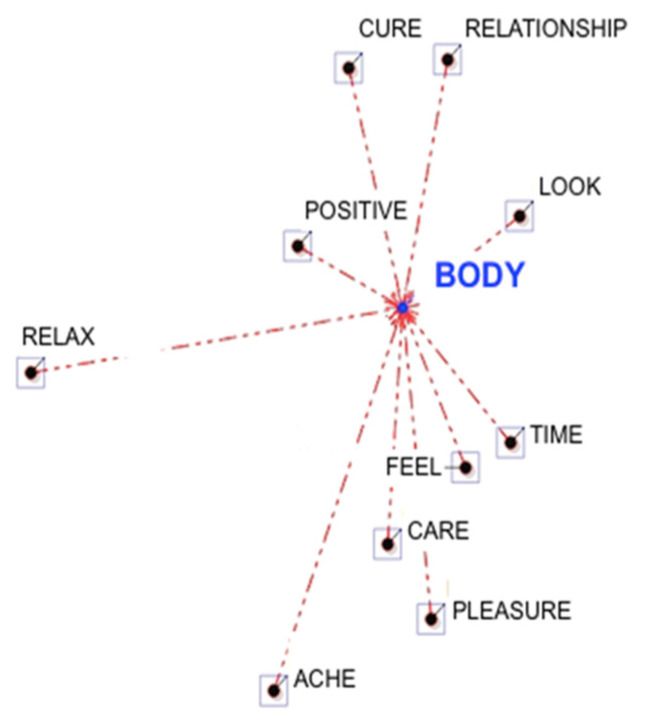
Graphical representation of the Word Association Analysis (after the psychological intervention).

**Table 1 ijerph-19-12515-t001:** Contents of the psychological intervention.

	Themes	Contents	Examples of Questions
**Session 1: Setting the Scene Motivation, Setting Goals for Change**	Introduction to the group and rules A **brief group “ice-breaker”** in which women introduce themselves and their reasons for attending the program Theoretical background on body image and self-compassion by the psychologist The **body image assessment**: participants filled a table focused to have a clear understanding of the problematic and maladaptive cognitive, emotional, and behavioral elements associated with body image before and after diagnosis and oncological treatments. Results were shared with each other into the group Participants defined specific goals on the basis of the results of the body image assessment Participants were invited to complete a **“Body Image Diary”** every day. The aim of this diary was to help survivors to monitor their body image experience focusing their attention to their body sensations, emotions, and behaviors.	Exploration of individual expectations A body image assessment Group motivational therapy	How did you feel about taking part in this project? What are your body image evaluations? What might be the benefits of attending to this project? What are your beliefs, emotions, and behaviors related to your body image?
**Session 2: The Survivors’ Relationship with their Body Image,** **Distorted Thinking and Core Beliefs on their Body Image**	Questions and feedback about the individual experience with the diary The construction of a **timeline of body image**: each participant wrote about barriers and facilitators in their relationship with the body every day. Moreover, they reflected on what predisposed to a negative body image by sharing their thoughts to the others in the group. The group discussed about the main related emotions and how practice self-compassion in a traumatic situation, such a breast cancer -The psychologist helps participants to relieve coping behaviors (e.g., avoidance, checking), which actually serve to worsen body image distress in the longer-term. Furthermore, specific behavioral strategies for improving positive behaviors towards their body were provided. These self-tailored strategies typically involve graduated exposure and response prevention interventions that patients develop within the group and set objectives for accomplishing them outside of the group. Finally, the psychologist instructed the patients to stay engaged in healthy activities (e.g., fitness and medications).	Exploring the impact of traumatic events on body image Understanding coping strategies and available safety behaviors Emphasis on thought’s recognition and possible positive consequences on everyday life	What are the main events linked to your body image experiences that you have had over the years? How do you react to these difficult experiences? How can you foster your safety behaviors?
**Session 3: Promoting Emotional Awareness and Self-Defeating Behaviors**	Questions and feedback about the individual experience with the diary The **“Body and Mind Relaxation”**: group members draw on paper their shape and label each of their body’ parts with an emotion. Then, the group discussion aimed at fostering their control on body image in relation to distress-provoking stimuli. In addition, the psychologist proposed various ways in which self-compassion may assist participants in overcoming negative emotions. Participants were encouraged to treat themselves with self-compassion when they dealt with their bodily issues -The psychologist proposed a ‘stop, look, and listen’ technique to identify inner sensations and cognitive distortions. Moreover, specific strategies for modifying cognitive distortions were provided. Patients were instructed to stay in contact with their inner sensations daily, incorporate cognitive restructuring exercises, and discover the emotional and behavioral positive consequences of changes. This way, patients were guided to identify dysfunctional body image-related schema (e.g., cognitive errors) and replaced a faulty self-talk with a new positive one.	Self-monitoring and listening of inner sensations were introduced	To what extent are you in contact with your thoughts and feelings? What happens to make you think or feel a certain way? What does emotional discomfort lead you to do? What do you believe about yourself? How vulnerable do you feel given existing societal pressures?
**Session 4: Reviewing Future Goals to Preserve a Positive Body Image and Support Self-Compassion Attitudes**	Questions and feedback about the individual experience with the diary -Participants **reviewed the individual and group** results and received feedback about attained changes by the conductor. They reflected on what vulnerabilities were activated and unfolded in day-to-day thoughts, emotions, and behaviors and how they learned to stop this self-perpetuating. In addition, participants reviewed their Body Image Diary and the relapse-prevention strategies to cope with future situations that might induce negative body image experiences. Finally, they set goals for further needed changes.	Review of the individual progresses and future goal setting	What have you learned about your body image? How can you manage body image distress, in the future?

**Table 2 ijerph-19-12515-t002:** Word association analysis before and after the psychological intervention.

Lemmas (Before the Intervention)	Coeff.	EC (B)	EC (AB)	*X^2^*	*p*
Seek	0.67	5	5	2.73	0.10
See	0.67	5	5	2.73	0.10
Feel	0.62	6	5	0.51	0.48
Unease	0.60	4	4	1.98	0.16
Energy	0.60	4	4	1.98	0.16
Pleasure	0.60	4	4	1.98	0.16
to be able to	0.60	4	4	1.98	0.16
Accept	0.52	3	3	1.36	0.24
Look	0.45	4	3	0.01	0.93
Physique	0.43	2	2	0.84	0.36
**Lemmas (After the Intervention)**	**Coeff.**	**EC (B)**	**EC (AB)**	** *X^2^* **	** *p* **
Feel	0.75	5	5	5.00	0.03
Care	0.67	4	4	3.64	0.06
Positive	0.67	4	4	3.64	0.06
Cure	0.60	5	4	1.25	0.26
Relationship	0.60	5	4	1.25	0.26
Look	0.58	3	3	2.50	0.11
Time	0.58	3	3	2.50	0.11
Pleasure	0.50	4	3	0.51	0.48
Ache	0.33	4	2	0.23	0.63
Relax	0.19	3	1	1.11	0.29

Note. The first ten lemmas with the higher coefficients were reported in the table. Coeff. = value of the coefficient; EC (B) = total amount of elementary context that contains every associated lemma (B); EC (AB) = total amount of elementary context where lemmas “A” and “B” are associated (co-occurrences).

**Table 3 ijerph-19-12515-t003:** Main themes and sub–themes emerged in the thematic analysis.

	Before the Psychological Intervention	*n*	After the Psychological Intervention	*n*
**Relationship with the Body**	1.1 *un-acceptance*: incapacity to accept the body changed after cancer	7	1.1 *un-acceptance*: incapacity to accept the body changed after cancer	3
	1.2 *distance from the body*: avoid looking and touching their bodies	1	1.2 *staying in contact with the body*	3
	1.3 *body as a cancer-related reminder*	1	1.3 *having confidence with the body*	3
	1.4 *body collaboration:* be aware of its efforts	2	1.4 body collaboration: the body role and relevance to sustain the overall well-being	7
	1.5 *an “insecure attachment”*	1	/	0
**Emotions**	2.1 *sense of alienation*: perception of a body that is not under control	1	2.1 *sense of alienation*: absence of the bodily sensations	1
	2.2 *unclear body signals*: the body gives information that are difficult to be understood	1	/	0
	2.3 *fear of cancer recurrence*: sensations are linked to the fear of a new cancer diagnosis	1	/	0
	2.4 *negative emotions*: rejection, discomfort, and contempt towards their body	5	2.4 *negative emotions*: anger, discomfort, sadness, and guilt	3
	2.5 *ambivalent emotions*: presence of both positive and negative feelings towards the body	4	2.5 *ambivalent emotions*: presence of both positive and negative feelings towards the body	4
	2.6 *sense of gratitude*: gratitude for the efforts endured by the body	3	2.6 *sense of gratitude*: gratitude for bearing oncological treatment; awareness related to the relevance of the total body	4
**Thoughts**	3.1 *cognitive burden*: fatigue related to the perception of a tired body and body rejection	4	3.1 *cognitive burden*: fatigue related to the perception of a tired body and body rejection	3
	3.2 *sense of impotence:* impossibility to do something to come back to their bodies before breast cancer	2	3.2 *sense of impotence:* impossibility to do something to come back to their bodies before breast cancer	1
	3.3 *take care and time:* listening of inner sensations and relaxation practice.	4	3.3 *take care and time*: be involved in relaxation practice; more awareness about the importance of the body to promote the overall well-being	6
	/	0	3.4 *the body as a helper:* body is now perceived as an ally	4

## Data Availability

Not applicable.

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
