# Peer review of "The Body after Cancer: A Qualitative Study on Breast Cancer Survivors’ Body Representation"

_ijerph, 2022, doi:10.3390/ijerph191912515_

Round 1
Reviewer 1 Report
In my opinion the aim of the study is extremely interesting. I would change the disscusion a little bit, maybe the authorscould disscuss their resutls with the data obtained by another authors. I couln't find the info on which stage of desease were thsese participants ( in another words: for how long have they been souvivors?) I really appreciate that the limitations of the study were discussed very carefully.
Author Response
Reviewer 1
In my opinion the aim of the study is extremely interesting. I would change the disscusion a little bit, maybe the authors could discuss their resutls with the data obtained by another authors. I couln't find the info on which stage of disease were thsese participants (In another words: for how long have they been souvivors?) I really appreciate that the limitations of the study were discussed very carefully.
Dear Reviewer,
Thank you for your work on our manuscript and for the positive comments. Starting from your suggestions, we improved this contribution as depicted below. Kindly notice that any modification to the manuscript has been highlighted in green to aid consultation of review.
- As suggested, the discussion has been enriched with other contributions in order to compare data obtained to the current literature. (e.g., Pearl et al., 2020; Kalaitzi et al., 2007; Breidenbach et al., 2022; Kneip et al., 2022)
References:
- Breidenbach, C., Heidkamp, P., Hiltrop, K., Pfaff, H., Enders, A., Ernstmann, N., & Kowalski, C. (2022). Prevalence and determinants of anxiety and depression in long-term breast cancer survivors. BMC psychiatry, 22(1), 1-10. https://doi.org/10.1186/s12888-022-03735-3
- Kalaitzi, C., Papadopoulos, V. P., Michas, K., Vlasis, K., Skandalakis, P., & Filippou, D. (2007). Combined brief psychosexual intervention after mastectomy: Effects on sexuality, body image, and psychological well‐Journal of surgical oncology, 96(3), 235-240. https://doi.org/10.1002/jso.20811
- Kneip Pelster, A. D., Coleman, J. D., Jawed-Wessel, S., Irwin, J. A., Heerten-Rodriguez, L., & Fisher, C. M. (2022). Sexuality, Breast Cancer Survivorship, and Script Theory. Sexuality Research and Social Policy, 1-10. https://doi.org/10.1007/s13178-021-00672-w
- Pearl, R. L., Wadden, T. A., Bach, C., Gruber, K., Leonard, S., Walsh, O. A., ... & Berkowitz, R. I. (2020). Effects of a cognitive-behavioral intervention targeting weight stigma: A randomized controlled trial. Journal of consulting and clinical psychology, 88(5), 470. https://doi.org/10.1037/ccp0000480
2) Second, we added Inclusion and Exclusion criteria in order to clarify the sample’s characteristics. In particular, participants are breast cancer survivors who are not under oncological treatment currently.
3)Thank you for your appreciation. Doing the revisions, the limitations were further enriched by promoting the information needed and adding other considerations. In particular, limitations about results generalizability, the sample size and characteristics were added.

Reviewer 2 Report
I have read the manuscript entitled: The Body After Cancer: A Qualitative Study on Breast Cancer Survivors’ Body Representation. In this study, Sebri V. et al aimed to qualitatively explore the bodily representations on a cognitive and emotional level in breast cancer survivors. A qualitative lexicometric analysis and a thematic analysis were carried out to assess the relationship with their body and related emotions and thoughts in an online psychological intervention focused on self-compassion. This study presents changes in the description of the relationship with their bodies as well as participants’ emotions and thoughts before and after a psychological intervention. This study is the first investigating the specific changes in bodily representations of breast cancer survivors, after an intervention which focused on self-compassion towards their body and was delivered in an online setting.
The manuscript is clearly written. The introduction provides sufficient background. The study is well conducted, the results are sufficiently described, but table 1 is not necessary. The authors should transfer Table 1 to supplementary materials. The conclusions of the authors are supported by the data described.
The authors have done well in acknowledging facts in the "Limitations" and "Conclusion" section. I agree with the authors that did not include a control group nor a quantitative evaluation of the intervention efficacy with validated measures. The sample size is too low, and needs to be increased for this sort of studies. In the abstract, the number of respondents is 15, while in the results 18.
The Materials and Methods should be described with sufficient details to allow others to replicate and build on the published results. In this section is no information when the research was carried out? During the COVID-19 pandemic? Did women with mental disorders take part in the research? Do the authors know if the patients were taking any medications that could affect their well-being?
By citing various studies in their discussion, the authors have highlighted the differences in study methods and their impact on the outcomes. For example, it would be great if the authors could discuss some of the controversy around accepting the body after cancer. Moreover, there should be a more thorough discussion of previous work in this area.
References should be numbered in order of appearance in the text (including citations in tables and legends) and listed individually at the end of the manuscript and indicated by a numeral or numerals in square brackets—e.g., [1] or [2,3], or [4–6]. For embedded citations in the text with pagination, use both parentheses and brackets to indicate the reference number and page numbers; for example [5] (p. 10). or [6] (pp. 101–105).
The authors should describe the appropriate ethics committee. An approval from the local institutional review board (IRB) or other appropriate ethics committee must be obtained before undertaking the research to confirm the study meets national and international guidelines. As a minimum, a statement including the project identification code, date of approval, and name of the ethics committee or institutional review board must be stated in Section ‘Institutional Review Board Statement’ of the article.
Thank you
Author Response
Reviewer 2
I have read the manuscript entitled: The Body After Cancer: A Qualitative Study on Breast Cancer Survivors’ Body Representation. In this study, Sebri V. et al aimed to qualitatively explore the bodily representations on a cognitive and emotional level in breast cancer survivors. A qualitative lexicometric analysis and a thematic analysis were carried out to assess the relationship with their body and related emotions and thoughts in an online psychological intervention focused on self-compassion. This study presents changes in the description of the relationship with their bodies as well as participants’ emotions and thoughts before and after a psychological intervention. This study is the first investigating the specific changes in bodily representations of breast cancer survivors, after an intervention which focused on self-compassion towards their body and was delivered in an online setting.
The manuscript is clearly written. The introduction provides sufficient background. The study is well conducted, the results are sufficiently described, but table 1 is not necessary. The authors should transfer Table 1 to supplementary materials. The conclusions of the authors are supported by the data described.
The authors have done well in acknowledging facts in the "Limitations" and "Conclusion" section. I agree with the authors that did not include a control group nor a quantitative evaluation of the intervention efficacy with validated measures. The sample size is too low and needs to be increased for this sort of studies. In the abstract, the number of respondents is 15, while in the results 18.
The Materials and Methods should be described with sufficient details to allow others to replicate and build on the published results. In this section is no information when the research was carried out? During the COVID-19 pandemic? Did women with mental disorders take part in the research? Do the authors know if the patients were taking any medications that could affect their well-being?
By citing various studies in their discussion, the authors have highlighted the differences in study methods and their impact on the outcomes. For example, it would be great if the authors could discuss some of the controversy around accepting the body after cancer. Moreover, there should be a more thorough discussion of previous work in this area.
References should be numbered in order of appearance in the text (including citations in tables and legends) and listed individually at the end of the manuscript and indicated by a numeral or numerals in square brackets—e.g., [1] or [2,3], or [4–6]. For embedded citations in the text with pagination, use both parentheses and brackets to indicate the reference number and page numbers; for example [5] (p. 10). or [6] (pp. 101–105).
The authors should describe the appropriate ethics committee. An approval from the local institutional review board (IRB) or other appropriate ethics committee must be obtained before undertaking the research to confirm the study meets national and international guidelines. As a minimum, a statement including the project identification code, date of approval, and name of the ethics committee or institutional review board must be stated in Section ‘Institutional Review Board Statement’ of the article.
Thank you
Dear Reviewer,
Thank you for your time and precious suggestions on our Manuscript. We hope that the revisions improved the quality of the manuscript. Please note that any modifications to the manuscript have been highlighted in green. Please, see detailed responses below and in the new version of the manuscript. In addition, we adjusted the Manuscript as follows:
1) Table 1 has been removed and transferred as Supplementary Material, as you suggested.
2) We checked the number of respondents.
3) Materials and Methods section was enriched with the details suggested. Moreover, we added “March, 2022” as the month in which the psychological intervention was conducted. Inclusion and exclusion criteria were also presented in order to clarify that women with mental impairments and under oncological treatments were excluded. Thank you for this comment.
4) Discussion was enriched with other contributions to enrich knowledge about data obtained and give a useful contribution to the current literature.
5) The references were adjusted following the reviewers’ comments and the journal guidelines.
6) Following this suggestion, the “Institutional Review Board Statement” was added in the Methods and Materials sections, before Participants.

Reviewer 3 Report
great work
I just added comments in the attached manuscript
to clarify some minor details
very interesting

Author Response
Reviewer 3
great work.
I just added comments in the attached manuscript
to clarify some minor details
very interesting
Dear Reviewer,
thank you for your positive evaluation and your efforts to promote our Manuscript. We adjusted the Manuscript following your suggestions. Please note that any modifications to the manuscript have been highlighted in green. Thank you again for the time you dedicated to our contribution

Round 2
Reviewer 2 Report
Thank you for the corrections made. In this study, Sebri V. et al aimed to qualitatively explore the bodily representations on a cognitive and emotional level in breast cancer survivors. This study is the first investigating the specific changes in bodily representations of breast cancer survivors, after an intervention which focused on self-compassion towards their body and was delivered in an online setting.The manuscript is clearly written. The introduction provides sufficient background. The study is well conducted, the results are sufficiently described. The discussion was completed.